# A Randomized Clinical Trial to Assess the Efficacy of Online-Treatment with Trial-Based Cognitive Therapy, Mindfulness-Based Health Promotion and Positive Psychotherapy for Post-Traumatic Stress Disorder during the COVID-19 Pandemic: A Study Protocol

**DOI:** 10.3390/ijerph19020819

**Published:** 2022-01-12

**Authors:** Érica Panzani Duran, Curt Hemanny, Renata Vieira, Orlando Nascimento, Leonardo Machado, Irismar Reis de Oliveira, Marcelo Demarzo

**Affiliations:** 1Postgraduate Program of Interactive Processes of Organs and Systems, Health Sciences Institute, Department of Neuroscience and Mental Health, Federal University of Bahia, Salvador 40110-060, Brazil; hemanny@gmail.com (C.H.); pesquisa.tept.2020@gmail.com (R.V.); irismar.oliveira@me.com (I.R.d.O.); 2Mente Aberta—Brazilian Center for Mindfulness and Health Promotion, Department of Preventive Medicine, Universidade Federal de São Paulo (UNIFESP), Sao Paulo 04753-060, Brazil; orlando.ulisses@hotmail.com (O.N.); demarzo@unifesp.br (M.D.); 3Postgraduate Program in Neuropsychiatry and Behavioral Sciences, Center for Medical Sciences, Department of Neuropsychiatry, Federal University of Pernambuco (POSNEURO-CCM-UFPE), Recife 50070-460, Brazil; leonardo.machadot@ufpe.br

**Keywords:** post-traumatic stress disorder, trial-based cognitive therapy, positive psychotherapy, mindfulness-based health promotion, COVID-19

## Abstract

Background: Research suggests the use of different forms of therapy as a way of decreasing dropout rates in the treatment of post-traumatic stress disorder (PTSD). The psychotherapies to be assessed in this study are trial-based cognitive therapy (TBCT), mindfulness-based health promotion (MBHP) and positive psychotherapy (PPT). Objectives: (1) to assess the online efficacy of TBCT compared to MBHP and PPT to reduce the symptoms of PTSD in the context of the Coronavirus Disease 2019 (COVID-19) pandemic; (2) to compare the efficacy of these psychotherapies in improving anxiety, depression, guilt and in promoting well-being; and (3) to describe how professionals perceive online treatment. Methods: A randomized, multicenter, single-blind clinical trial will be conducted, with three separate arms. An estimated sample of 135 patients will receive either TBCT, MBHP or PPT and will be treated through online, individual, weekly visits, totaling 14 sessions. The primary outcome will be CAPS-5 and secondary outcomes will be HADS and WHO-5. The variables used to mediate these outcomes will be the Trauma-Related Guilt Inventory (TRGI), Negative Core Beliefs Inventory (NCBI) and the California Psychotherapy Alliance Scale (CALPAS-P). Expected results: PTSD symptoms are expected to be reduced after TBCT, MBHP and PPT. No statistical difference is expected to be found among the three. Discussion: The present study will evaluate and contribute towards the development of new psychotherapeutic options for patients with PTSD. The results of this study will allow the dissemination of new effective and adaptable interventions for patients with PTSD.

## 1. Introduction

According to the American Psychiatric Association (DSM-5) [1], post-traumatic stress disorder (PTSD) occurs after exposure to one or more traumatic events, including actual or threatened death, serious injury or sexual violence in situations such as direct experience of the traumatic event, witnessing as it occurs to others or learning about a violent or accidental traumatic event that happened to a family member or close friend. Furthermore, in accordance with the DSM-5, experiencing repeated or extreme exposure to aversive details of the traumatic events may provoke PTSD.

Although exposure to traumatic events is a global problem that carries the risk of producing PTSD [2], the vast majority of people have a brief acute stress response. Around 80% of people recover in the first month after the event without long-term consequences [3]. While epidemiological studies indicate prevalence rates ranging from 3.8% to 8.3% in the general population [4], this prevalence varies from 14.8% to 83% in people working during emergencies, such as physicians and nurses [5,6,7].

With the COVID-19 global crisis, the literature and experience in health care indicate that lasting emotional trauma can arise on an unprecedented scale [8], with devastating consequences such as unemployment, deaths and social isolation. Exposure to a serious illness leads to feelings of vulnerability, fear, panic and despair. No previous catastrophic events may be compared to the COVID-19 pandemic in terms of geographical scale [9]. World Wars I and II and other pandemics, for example, were geographically limited, saving some continents [10]. The Spanish flu, with high mortality rates in the first decades of the twentieth century, did not disseminate on a global scale. Furthermore, there is limited literature to help people deal with traumatic events, such as quarantine, mass disaster and ongoing stressors, making the damage to mental health on a large scale seem imminent. People who are quarantined are very likely to develop a wide range of symptoms of psychological stress and disorder, including depression and post-traumatic stress symptoms [11].

Thus, it is pertinent to look for effective psychotherapeutic interventions for people with PTSD that can be implemented online, facilitating the immediate availability of mental health care, regardless of the geographic region. Because of the empirical support that cognitive behavioral therapy (CBT) [12,13,14,15] and prolonged exposure therapy (PE) [16,17,18] have received, they are currently the first-line treatments for PTSD [19].

Available PTSD treatments have been shown to have a high non-response and dropout rate. A review of the literature of 55 studies of PTSD patients found non-response rates as high as 50% and dropout rates ranging widely and depending on the nature of the study population [20]. In a meta-analysis assessing war trauma [21], dropout rates ranged from 5% to 78%, and in a recent review of data from three clinical trials of evidence-based PTSD treatments in military service members, the dropout rate was 31% [22]. These findings suggest new and innovative strategies are needed to improve treatment retention, especially in a new and complex situation such as the COVID-19 pandemic. Thus, testing treatments for PTSD with lower potential for dropouts might still be necessary [23,24]. Recently, Duran et al. [25] showed that TBCT had a low dropout rate in PTSD relative to PE.

TBCT [26,27,28] is a novel and innovative transdiagnostic approach [29] shown to be effective for depression [30], social anxiety disorder [31,32,33,34], PTSD [25] and obsessive-compulsive disorder (Rodrigues et al., accepted). TBCT differs from other CBT approaches in that it introduces a new, organized and systematic approach to change dysfunctional negative core beliefs (CBs), while allowing cognitive, emotional and experiential work to be done conjointly [28]. While incorporating a courtroom metaphor to challenge dysfunctional CBs conceptualized as self-accusations, TBCT is an example of assimilative psychotherapy integration that relies on Beckian CBT, incorporating and integrating components of other psychotherapies [28,35].

The Mindfulness-Based Stress Reduction (MBSR) program was created by Jon Kabat-Zinn and colleagues at the University of Massachusetts Medical Center in 1979, and it is an intervention whose effects on mental health and quality of life have produced several studies worldwide, both in clinical and non-clinical populations. Several protocols have been developed based on the MBSR aimed at specific populations, such as the Mindfulness-Based Health Promotion (MBHP) program developed by the Mente Aberta Brazilian Center for Mindfulness and Health Promotion. The MBHP program was inspired by the original MBSR model but adapted to the context of the Brazilian health care (SUS) system, addressing chronic conditions and mental disorders as well [36,37,38].

Positive psychology seeks to understand positive emotions, psychological potentialities and healthy human/social/institutional functioning, and apply this knowledge to help people and institutions, with a focus on the prevention and promotion of mental health [39]. At first, positive psychology focused a lot on happiness and subjective well-being [40,41]. In a second step, the studies gained a broader view of psychological well-being and another similar proposal entitled PERMA, which is composed of the following five spheres: *positive emotions*—*P; engagement*—*E; relationships*—*R; meaning*—*M; achievement*—*A* [42,43]. Although positive psychology aims to be a way of looking at life, some psychotherapeutic proposals, such as positive psychotherapy (PPT), have been developed, and clinical studies have been replicated in different clinical and cultural contexts [44,45].

The proposed study is a randomized control trial comparing TBCT, MBHP and PPT to treat patients with PTSD due to traumatic exposure during the COVID-19 pandemic. Our hypotheses are that TBCT is at least as effective as MBHP and possibly more effective than PPT, considering that no studies were found assessing its efficacy in patients with PTSD.

Our primary objectives are to assess the efficacy of TBCT for patients with PTSD because of or during the COVID-19 pandemic and to compare its efficacy with that of MBHP and PPT, with all therapies delivered online in the same sample. Secondarily, we aim to quantitatively assess symptoms of depression, anxiety and guilt, as well as well-being and the strength of the therapeutic alliance.

## 2. Materials and Methods

### 2.1. Study Design

This study project is a randomized single-blind control trial with three parallel groups. After sample selection, patients with PTSD that developed or recurred as a consequence of the COVID-19 pandemic will be randomly assigned to TBCT, MBHP or PPT, with all therapies delivered online. Figure 1 shows the flowchart of the study. The duration of treatment will be 14 to 17 weeks, with a follow-up of 3, 6 and 12 months. The online psychotherapies will be carried out individually and weekly, totaling 14 one-hour sessions. Outcome measures will take place before session 1, session 7 and session 14, as well as at 3, 6 and 12 months. This study protocol was written following the standard protocol items, recommendations for interventional trials (SPIRIT) [46], has been registered on Clinicaltrials.gov (NCT04852770) and approved by the Ethics Committee of Maternidade Climério de Oliveira at Federal University of Bahia, Brazil. This study was funded by the National Council for Scientific and Technological Development (CNPq).

### 2.2. Hypotheses

According to the literature and the results of other clinical trials, we have elaborated the following hypotheses:

**Hypotheses 1** **(H1)**.*TBCT and MBHP in online modalities are statistically different from PPT online in reducing the symptoms of PTSD*;

**Hypotheses 2** **(H2)**.
*TBCT and MBHP in online modalities are of greater effect size in reducing PTSD symptoms compared to the control (PPT);*


**Hypotheses 3** **(H3)**.*TBCT and MBHP in online modalities contain the largest effect sizes in reducing the symptoms of guilt over the traumatic event compared to the control*;

**Hypotheses 4** **(H4)**.*TBCT and MBHP in online modalities are statistically different from the control, with a larger effect size, in improving quality of life (or disability)*.

### 2.3. Participants

After announcements of the study on the media, interested people seeking online care as a result of COVID-19 will be selected through a self-administered questionnaire, the PTSD Checklist for DSM-5 (PCL-5), which contains a list of PTSD symptoms [47,48,49]. Patients will be contacted by trained evaluators who will use the Structured Clinical Interview for DSM-5 (SCID-5) to ensure the presence of the diagnosis according to DSM-5 criteria. Participants will receive explanations about the study and how psychotherapy works. Those agreeing to participate will sign the informed consent, which includes permission for sessions to be audiotaped. After completion of the self-assessment questionnaires, patients will be randomized to one of the study arms.

### 2.4. Inclusion Criteria

Subjects eligible to participate in this study are adults of both sexes, aged 18 to 60 years old and scoring 45 or more on the PCL-5 as a result of direct or indirect exposure to COVID-19 (e.g., health professionals, people who previously tested positive for COVID-19, or those who quarantined or isolated themselves). Participants should be able to read, write and follow instructions, as well as have access to the Internet.

### 2.5. Exclusion Criteria

Candidates to participate in this study who meet any of the following criteria will be excluded: severe suicide risk (plans, attitudes or suicide attempts during the last 12 months); self-mutilation behavior during the last 12 months; already engaged in another psychotherapy; presenting with psychotic symptoms; and substance use disorder in the last 12 months.

### 2.6. Study Settings

Eligible patients may be located anywhere in the Brazilian territory. The assessors and psychotherapists live in the following Brazilian cities: Nova Iguaçu, Recife, Rio de Janeiro, Salvador São Paulo and Uberlandia. The Federal University of Bahia, in the city of Salvador, is the study headquarters and also home to the principal investigators.

### 2.7. Interventions

The three psychotherapeutic group interventions will be delivered through Skype or Google meeting platforms as weekly, one-hour long sessions, totaling 14 sessions (14 to 17 weeks). The sessions will be audiotaped in order to assure reliability in the fidelity of the intervention and to allow review by qualified supervisors.

Trial-based cognitive therapy

TBCT is a psychotherapeutic intervention whose foundation lies on Beckian CBT [28,50]. Although TBCT, like CBT, includes psychoeducation, cognitive restructuring and behavior experiments assigned as homework, it differs from other CBT approaches in terms of formulation (e.g., TBCT conceptualization diagram), techniques (e.g., participation grid to deal with guilt) and measurement instruments, such as the Cognitive Distortions Questionnaire (CD-Quest) [51] and the Negative Core Beliefs Inventory (NCBI) [52], making TBCT a distinctive approach in restructuring cognition at the three levels of cognition, especially in modifying the dysfunctional negative CBs of patients.

The main TBCT technique, The Trial, was inspired by Kafka’s novel “The Trial” [53]. It was designed to challenge and restructure dysfunctional CBs. A parallel with the CBT perspective is proposed by de Oliveira [54], who argues that the self-accusations presented by the novel’s character, Joseph K., might correspond to the dysfunctional negative CBs about the self. Like Joseph K., patients are not aware of these CBs/self-accusations, but after an investigation by means of the downward arrow approach, the patients become aware of such CBs/self-accusations and learn how to build a proper defense for themselves [54]. The Trial technique includes several CBT components (e.g., examining the evidence) and the empty-chair procedure [55,56], which makes TBCT more experiential [57].

Several studies support the efficacy of TBCT. In the first and preliminary study (*N* = 30) with patients presenting with different diagnoses, this treatment effectively decreased the patients’ attachments to dysfunctional negative CBs, as well as the emotional intensity, during a session [26]. This study [57] was replicated with 166 patients with varying diagnoses and comorbidities, showing a significant reduction in the attachment to dysfunctional CBs and in emotional intensity.

It has also been demonstrated in a randomized clinical trial (*N* = 36) comparing TBCT techniques to conventional CBT techniques that TBCT was as efficacious as conventional CBT techniques in reducing social anxiety symptoms and improving quality of life and more efficacious than CBT in reducing fear of negative evaluation, social avoidance and distress [31,34]. A second study aiming to evaluate the efficacy of TBCT for generalized social anxiety disorder in a population with high rates of comorbid disorders, especially depression [33], was carried out. This was a two-arm randomized clinical trial that included 39 adults (TBCT = 18; waitlist group = 21), and symptom severity was assessed at pre- and post-treatment. Reductions in social anxiety, social avoidance and depression were observed in the participants in the TBCT group, of which all were associated with a large effect size, while no differences between pre- and post-treatment scores were observed in the waitlist condition. Interestingly, patients with comorbid conditions showed greater reductions in social anxiety symptoms across treatment relative to those with SAD only.

A study (*N* = 76) comparing the efficacy of TBCT, behavioral activation (BA) and treatment as usual (TAU) in the treatment of major depressive disorder [30] showed that both TBCT and BA (which also included antidepressants) were different from TAU (which included antidepressants alone) in reducing the HAM-D and BDI scores.

Recently, 95 patients who met DSM-4-TR criteria for PTSD were randomly assigned to receive either TBCT or prolonged exposure (PE) [25]. A significant reduction in the primary outcome as assessed by the Davidson Trauma Scale (DTS) was improvement in PTSD symptoms. Reductions in DTS scores were observed in both arms, but there was no significant difference between treatments. Regarding the secondary outcomes (depression, anxiety and dysfunctional attitudes assessed by the Beck Depression/Anxiety Inventories and Dysfunctional Attitudes Scale, as well as the dropout rate), significant differences in depressive symptoms were observed in favor of TBCT. In addition, the dropout rate was lower in the TBCT group than in the PE group, suggesting that TBCT may be an effective alternative for treating PTSD.

Mindfulness-based health promotion

“*Mindfulness-Based Health Promotion (MBHP)”* is the *Mindfulness* program protocol adopted by the Mente Aberta Center and has therefore become the basis of its professional training. MBHP was inspired by Jon Kabat-Zinn’s original model, *Mindfulness-Based Stress Reduction*, adapted to the context of Health and Quality of Life Promotion. Besides mindfulness-based stress reduction (MBSB), other inspirations have contributed to the MBHP protocol, including *Mindfulness-Based Cognitive Therapy* (MBCT) and *Mindfulness-Based Relapse Prevention*, both programs used by British Institutes [58,59].

The MBHP program was designed for the Brazilian and Hispanic-American context and in terms of its applicability in public policies in the areas of health, education and organizations. Potential beneficiaries of MBHP are public health care system professionals and patients (particularly those with chronic transdiagnostic conditions), public school students and teachers, police officers and other public workers, as well as the population at large. Evidence on the MBHP protocol has accumulated over the last decade, showing its efficacy and effectiveness in improving health-related quality of life, anxiety, depression and burnout symptoms; in particular through mechanisms such as decentering and self-compassion [36,37,38]. There are no studies specifically on PTSD symptoms, and this project addresses this gap. On the other hand, other mindfulness protocols have been shown to be effective for PTSD, especially those inspired by MBHP, such as the MBSR and MBCT [58,59].

Just as is the case with the classical *Mindfulness* approaches (MBSR and MBCT), the purpose of MBHP is to develop “Awareness” through the practice of *Mindfulness* (which involves attention, attitude and intention). Awareness is understood as being conscious (*aware*) of (realizing, acknowledging, perceiving, noticing and observing) inner phenomena (thoughts, feelings, emotions, sensations and impulses) and external phenomena (activities, relations, etc.). The assumption (principle/intention) is that, if the person is aware, he/she is more likely to make more assertive/conscious decisions/choices, responding to situations in a less reactive way (no more acting on “auto-pilot”). There is scientific evidence showing that developing *Awareness* is one of the basic mechanisms that explain the benefits of *Mindfulness* in health promotion, bringing self-efficacy and quality of life.

The definition of *Mindfulness* we use is that provided by *Jon Kabat-Zinn himself*: “*Mindfulness* is awareness that arises through paying attention, on purpose, in the present moment, non-judgmentally and non-critically, to the experience that emerges moment-by-moment.”

The original MBHP protocol is a structured program developed over eight sessions, in groups, where participants (8–15 people) meet every week for 2 h (standard duration of one session) to experience the concepts and techniques of mindfulness. However, MBHP was designed so it would allow customization when applied to public policies. If necessary, the number and duration of sessions may be changed, so it can be flexible and dynamic in its use by organizations, schools and health care services. Furthermore, to match the conditions of the present study, the protocol will be individual, 1 h sessions, delivered over 14 encounters (14 sessions).

Participants are also given suggestions of daily activities to be implemented at home or in the workplace, which last on average 15–20 min, but may last up to 45 min in the case of more motivated and compliant participants. They are also encouraged to incorporate the idea of *Mindfulness* in their daily lives (the so-called “informal practice”), so all daily activities somehow become opportunities to practice Mindfulness.

Positive psychotherapy

PPT is the clinical and therapeutic arm of positive psychology (PP). After empirical validation, PPT was organized in a cohesive protocol of 15 sessions. Many of the practices of PPT have been studied through online interventions. In the present study, the protocol will be reduced to 14 sessions, which is in agreement with the other arms of this clinical trial. PPT integrates symptoms with strengths, risks with resources, weaknesses with values and regrets with hopes, aiming to understand the inherent complexities of human experience in a balanced way. Without ignoring or minimizing the client’s concerns, the PPT clinician empathetically understands and pays attention to the pain associated with the trauma and simultaneously explores the potential for growth. PPT considers that clients in psychological distress can be better understood and served if they learn to use their highest resources—personal and interpersonal—to face the challenges in life [39,43,44,60].

PPT is divided into three phases. In phase 1, the patient creates a personal narrative, remembering and writing a story that awakened his/her best, especially in overcoming a challenge (a strategy called positive presentation). In subsequent sessions, which correspond to most of the therapeutic work in this phase, it focuses on the assessment and organization of a profile of the signature strengths and the acquisition of the skills necessary to integrate the forces with psychological stressors [44].

In phase 2, after establishing therapeutic rapport and helping clients identify their strengths, clinicians encourage them to write about their grudges, bitter memories or resentments and then discuss the effects of holding on to these negative aspects. PPT does not discourage the expression of negative expressions; it encourages clients to evaluate a wide range of emotions, positive and negative [44].

Phase 3 focuses on restoring or encouraging positive relationships. At this point, patients are likely to be ready to seek meaning and purpose; the forces have broadened the self-concepts of clients, and they are able to deal with disturbing memories, learn about forgiveness and begin to see the benefits of gratitude. Phase 3 also encourages patients to cultivate meaning through engaging in a number of processes, such as strengthening intimate and community personal relationships; the search for artistic, intellectual or scientific innovations; or engaging in philosophical or religious contemplation [44].

In his book “Flourishing”, Martin Seligman describes some programs applied in the US Army based on positive psychology. The results appeared to be promising, despite initial concerns [45]. However, other more detailed studies with a more rigid scientific design seem to be necessary. In any case, it is important to understand that the main focus of interventions based on positive psychology is not directly reducing symptoms of illness, that is, PTSD, but investing in aspects related to resilience, in this case, post-traumatic growth (PTG). PTG is defined as positive, meaningful psychological changes that an individual can experience as a result of coping with traumatic life events [61,62]. A review of nine qualitative studies with military and former military personnel showed that PTG is a phenomenon that can also occur after exposure to trauma in wars [46,62].

### 2.8. Outcomes

Primary outcomes will be assessed by the reduction in PTSD symptoms, measured through CAPS-5. Secondary outcomes will be the improvement in symptoms of anxiety, depression and well-being. Mediator variables will also be assessed, measuring negative CBs, guilt and the strength of the therapeutic alliance. All scales will be applied by a blind assessor to the intervention groups before randomization (Initial Assessment), after seven sessions (Intermediate Assessment) and after 14 sessions (Final Assessment). They will also be applied in the 3-, 6- and 12-month follow-ups. Table 1 shows the flow of sessions and evaluations over the study period. Table 1 shows the SPIRIT diagram.

### 2.9. Instruments

Diagnostic assessment

Structural Clinical Interview for the DSM-5 (SCID-5) [63]: It is a semi-structured psychiatric interview, with the purpose of providing a diagnosis, according to the Diagnostic and Statistical Manual of Mental Disorders (DSM-5).

*PCL-5 PTSD Check List 5:* It contains 20 items with the purpose of assessing PTSD symptoms, in accordance with criteria B, C, D and E (DSM-5). Each item is scored on a scale that ranges from 0 (not at all) to 4 (extremely) [47,48,49]. This scale will be applied only at intake.

Primary outcome

Clinician-Administered PTSD Scale (CAPS-5) [64]: It is a diagnostic interview scale with 30 items to assess the diagnosis and severity of PTSD symptoms in accordance with DSM-5 [65,66]. CAPS-5 will be applied at baseline, mid-treatment (session seven) and post-treatment (session fourteen), as well as at follow-up (3, 6 and 12 months).

Secondary outcome

*Hospital Anxiety and Depression Scale* (HADS) [65]: This is a 14-item scale used to determine anxiety and depression symptoms, where each item is worth 0 to 3 points, and a total score of 9 or higher suggests mild symptoms. It was translated and validated for the Brazilian population [67,68]. This scale will be administered at baseline (initial evaluation), mid-treatment (week 7) and post-treatment (week 14), as well as at 3-, 6- and 12-month follow-ups.

*The World Health Organization Five Well-Being Index (WHO-5):* This is an overall well-being scale, with five questions and scores ranging from 0 to 5, addressing mood and energy [69,70]. The WHO-5 validation study into Brazilian Portuguese included 1128 individuals. During this research, the instrument presented good internal validity (Cronbach’s alpha = 0.83) [71]. WHO-5 will be applied at baseline, mid-treatment (session seven) and post-treatment (session 14), as well as at the follow-ups (3, 6 and 12 months).

Mediator variables

Trauma-Related Guilt Inventory (TRGI) [72]: This is a 32-item questionnaire assessing the cognitive and emotional aspects of guilt that are associated to a specific traumatic event. TRGI will be applied at baseline, mid-treatment (session 7) and post-treatment (session 14), as well as at the follow-ups (3, 6 and 12 months).

Negative Core Beliefs Inventory (NCBI) [52]: This inventory is designed to assess negative CBs, as described by [73]. It consists of 50 items evaluating beliefs about oneself and the world, on a Likert scale of 1 to 4 points each. NCBI will be applied at baseline, mid-treatment (session seven) and post-treatment (session 14), as well as at the follow-ups (3, 6 and 12 months).

*California Psychotherapeutic Alliance Scale—patient version (CALPAS-P):* Scale that assesses four dimensions of the therapeutic relationship [74,75]: (1) therapeutic alliance, (2) working alliance, (3) understanding and (4) therapist involvement, based on an agreement between patient and therapist about the goals of therapy. The CALPAS-P has 24 items ranging from 1 (never) to 7 (always) on the Likert scale. This scale will be administered at the end of second, fourth, sixth, eighth, tenth, twelfth and fourteenth psychotherapy sessions.

### 2.10. Sample Size

The sample size calculation was estimated using the G*Power software version 3.1 [76], considering a mixed design. Parameters that were taken into account were the F test and repeated measures ANOVA for comparison between groups, for an effect size of 0.25, α criterion of 0.05 and power of (1−β) 0.80, three groups design (TBCT X MBHP X WBT), three measures (baseline, mid-treatment and final assessment), and an a priori correlation between the repeated measures of 0.50. Hence, a sample of 108 participants was obtained, with a numerator of 2 and a denominator of 105 degrees of freedom, for a critical F of 3.08. Considering a 25% dropout rate, a total of 135 participants was estimated.

### 2.11. Randomization and Sequence Generation

After screening and initial evaluation, the study coordinator will refer the patient to the person responsible for randomization, who does not participate in the patient treatment. The randomization process, at www.random.org (accessed on 1 December 2021), will be done through a computer-generated table of random numbers, allocating patients to a particular group. After that, the patient will be referred to the psychotherapist.

Six evaluators, blinded to the intervention group, will be responsible for assessing and referring the eligible patient to the study coordinator, who will proceed with the randomization. Each evaluator will assess the same patients throughout the study.

### 2.12. Implementation

After the patient has signed the informed consent and has been evaluated for inclusion criteria and baseline measurements, the assessor will contact the research coordinators informing them about the new patient and providing their patient number. Based on the random list, the coordinators will then contact the therapist to inform him/her about the new patient now available for treatment.

### 2.13. Randomization and Concealment

Patients who meet all eligibility criteria and complete the baseline assessments will be randomly allocated (1:1:1) to TBCT, MBHP or PPT. Randomization is based on a true random number service (http://www.random.org, accessed on 01 December 2021) and will be conducted by a member of the research team who is blinded to patient characteristics and is not involved in any other aspects of the study. The study therapists will be blinded to assessment outcomes, and the research assistant conducting post- and follow-up assessments will be blinded to participant condition.

### 2.14. Interventions Similarities and Differences

The interventions have similarities and differences. First of all, TBCT aims to identify thoughts, emotions and behaviors on three levels of cognition, with the purpose of changing them especially at the third level, where it is meant to identify and modify CBs, which mediate the change of thoughts and behaviors characterized as symptoms. On the other hand, MBHP and PPT are not meant to identify distorted cognitions, unpleasant emotions and dysfunctional behaviors or change them directly. MBHP, for instance, teaches from the very first sessions how to pay attention to the present moment and how to use various mindfulness practices. It teaches the patient how to identify rewarding characteristics and valuable personal attributes, without correlating these to pathological thoughts, emotions or behaviors. On the other hand, the focus of PPT is to enhance personal strengths; behaviors related to gratitude, kindness and appreciation; and thoughts related to well-being. Therefore, each intervention advocates a different change process: TBCT modifies CBs, MBHP teaches meditation skills and PPT guides the patient into enhancing personal strengths and virtues.

### 2.15. Statistical Methods

The collected data will be analyzed statistically using the Statistics Package for Social Science software (SPSS), version 24.0 [77]. Clinical and demographic categorical variables will be analyzed with *X*^2^ (sex, education, color or race, and dropout rate) and continuous variables will be assessed with one-way ANOVA (age and scale scores). Missing data for both primary and secondary outcome measures will be analyzed by intention to treat (ITT) and multiple imputation (IM), with five imputed databases taking baseline and sociodemographic data and assuming the random chance mechanism. Generalized estimation equations (GHG), taking time as a covariate, will be performed to compare changes in scores over time and between groups (TCP, MBSR and PP). Effect sizes will be evaluated according to Cohen’s tests. The assumed level of statistical significance will be 0.05.

### 2.16. Ethical Aspects

This study was approved by the Human Research Ethics Committee of the Maternity Hospital Climério de Oliveira, Federal University of Bahia (CAAE: 30769420.0.0000.5543) and registered on clinicaltrials.gov (NCT04852770).

## 3. Discussion

To the best of our knowledge, this is the first study at the international level to compare the effects of three different psychosocial interventions on PTSD symptons. Moreover, the current pandemic caused by the dissemination of the COVID-19 virus has caused worry and unprecedented impacts on mortality rates, on the economy and on mental health. The impact on the latter, however, has been very well established, as anxiety and depression symptoms, including suicidal behavior, have increased significantly in the general population as a result of the pandemic [78]. Symptoms are directly influenced by alarming, excessive and unreliable news; by the risks of contamination; by the risks of it affecting the individual’s own life or that of their loved ones; and by restriction on activities and financial losses. Long periods of uncertainty, insecurity and feelings of inefficacy bring about anguish and anxiety, as they are considered predictors of trauma. Studies suggest that individuals who are close to the most vulnerable patients tend to be more affected than health professionals on the frontline [78], because although the health care context involves higher risks, it also has more control measures against contamination. In any case, the general population is more vulnerable to stress- and trauma-related disorders, especially PTSD [79,80]. This is clearly a stress-related disorder with well-defined triggering cognitive models [81].

To date, psychotherapy is the first treatment option for PTSD, as it restructures trauma-related cognitions, thus reducing symptoms that impact quality of life, such as reactions of anxiety, stress and insomnia. In our study, from a randomized and controlled design, integrating quantitative and qualitative measures, and with a sample with good statistical power, we innovated by testing and comparing two new interventions that proved effective for mental disorders, mindfulness and positive psychology, which has the potential to bring relevant new clinical information to the international literature.

Qualitative data will be collected on how patients and psychotherapists see the pandemic as a whole and online therapies. Although data on the impact of the pandemic on health professionals have already been published [79], there are no qualitative studies addressing online psychotherapy during the pandemic.

Recently, a clinical trial indicated that TBCT is no different from prolonged exposure therapy in reducing symptoms, but presents higher rates of compliance, which suggests that this approach may be used to treat PTSD [25]. The same can be said about mindfulness-based psychotherapeutic interventions [82] and positive psychology [83], as both have proven to be promising in the reduction of trauma- and stress-related symptoms.

## 4. Conclusions

We hope this study will shed light on the impact of the online use of the psychotherapies, herein evaluated, in the context of the pandemic and for mental health in general. This study was specifically designed for the current times and in accordance with the health standards for social distancing. We believe it will help patients diagnosed with PTSD, which seems to be among the most prevalent disorders caused by COVID-19. Results will also help increase the body of evidence on the efficacy of TBCT, MBHP and PPT online, as these types of therapy are increasingly being used but still lack evidence.

## Figures and Tables

**Figure 1 ijerph-19-00819-f001:**
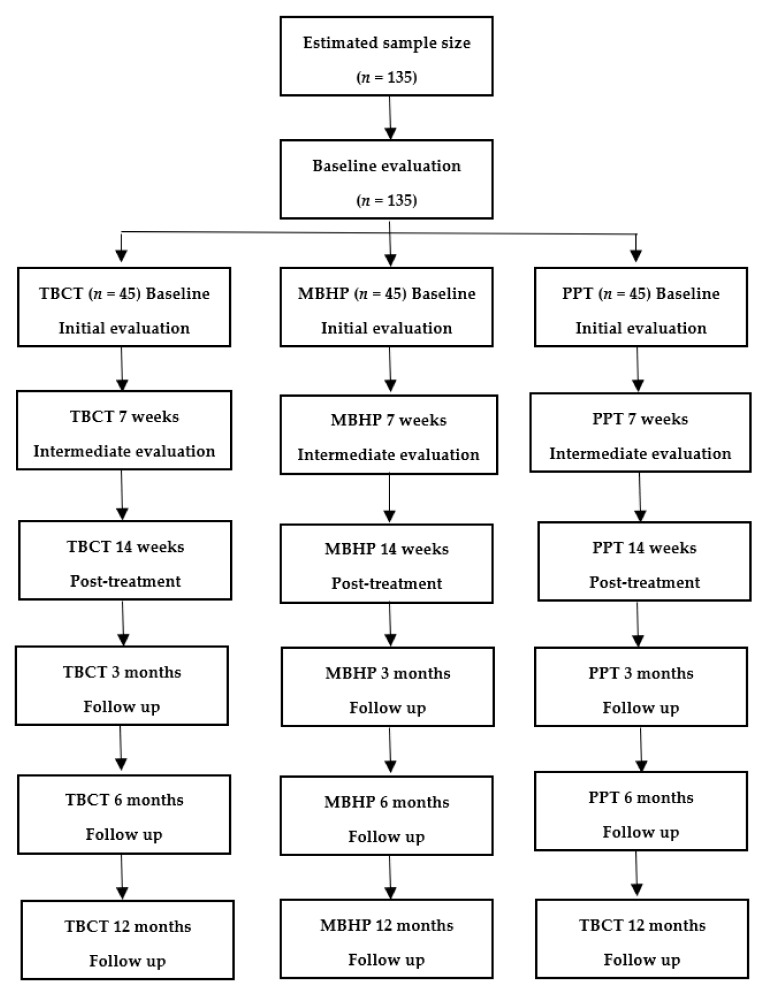
Flowchart of study (CONSORT) TBCT: trial-based cognitive therapy; MBHP: mindfulness-based health promotion; PPT: positive psychotherapy.

**Table 1 ijerph-19-00819-t001:** Flow of sessions and evaluations over the study period.

	Study Period
	Enrolment	Allocation	Post-Allocation	Follow-Up
**Timepoint**	Screening	Baseline	7 weeks	14 weeks	3 months	6 months	12 months
**Enrolment**:							
Eligibility screen	X						
Informed consent		X					
Allocation		X					
**Interventions**:							
TBCT			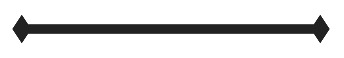			
MBHP			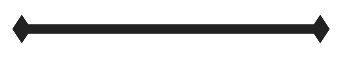			
PPT			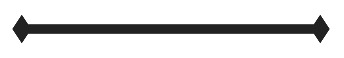			
**Assessments:**							
**PCL-5**	X						
**SCID-5**		X					
**CAPS-5**		X	X	X	X	X	X
**HADS**		X	X	X	X	X	X
**TRGI**		X	X	X	X	X	X
**NCBI**		X	X	X	X	X	X
**WHO-5**		X	X	X	X	X	X
**CALPAS-P**		X	X	X			
**SS interview**				X			

## Data Availability

The data presented in this study will be available on request from the corresponding author.

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
