# Peer review of "A Randomized Clinical Trial to Assess the Efficacy of Online-Treatment with Trial-Based Cognitive Therapy, Mindfulness-Based Health Promotion and Positive Psychotherapy for Post-Traumatic Stress Disorder during the COVID-19 Pandemic: A Study Protocol"

_ijerph, 2022, doi:10.3390/ijerph19020819_

Round 1
Reviewer 1 Report
The authors are to be commended for suggesting an online psychotherapy addressing posttraumatic stress disorder in the COVID-19 context. I have, however, the impression that too many questions are raised by a single study. It is true that your paper is well written and methodologically sound. But why do you study online psychotherapy AND compare three psychotherapies, given the dropout risk probably higher than 15 % (according to your literature research)? I wonder whether you should work on your research questions / hypotheses and simplify your design.
Author Response
Reviewer 1:
The authors are to be commended for suggesting an online psychotherapy addressing posttraumatic stress disorder in the COVID-19 context. I have, however, the impression that too many questions are raised by a single study. It is true that your paper is well written and methodologically sound. But why do you study online psychotherapy AND compare three psychotherapies, given the dropout risk probably higher than % (according to your literature research)? I wonder whether you should work on your research questions/hypotheses and simplify your design.
We thank Reviewer 1 for helping us improve the manuscript with his/her comments, and we hope our response is satisfactory.
Before designing this protocol, we reviewed previous studies on PTSD. We were impressed by Markowitz et al.’s (2015) study comparing three different forms of psychotherapy for PTSD: Interpersonal Psychotherapy, Prolonged Exposure (an exposure-based exemplar), and Relaxation Therapy (an active control psychotherapy).
Considering that Brazil was (and still is) one of the most affected countries by the COVID-19 pandemic, we thought that having as many available skilled therapists as possible to help people with PTSD all over the country would be a need. The first author then contacted several university research centers in the country with experience in treating PTSD with psychotherapy. Many were contacted, but only three research centers accepted to participate, namely, Federal University of São Paulo (Mindfulness-Based Health Promotion), Federal University of Pernambuco (Positive Psychotherapy), and Federal University of Bahia (Trial-Based Cognitive Therapy). The online format would give the opportunity to cover the whole country, reaching small cities and villages served by the internet.
Thus, besides backed by a previous example in the literature (Markowitz et al., 2015), the main reason for offering the three forms of psychotherapy online was pragmatic. As we are now threatened by a new variant and a fourth wave of the pandemic, which is apparently far from an end, mental health issues, and particularly PTSD, persist and will probably last long.
Dropout rates were a concern, reason why we conducted a small sample size pilot study (N=12), with 4 patients per arm. Dropout rates were in the expected range (33% per arm).
We agree with Reviewer 1 that our design could be simplified. However, considering that this protocol has already been approved by three local IRDs, and by the national ethics committee, we are afraid changing our design would take many more months, delaying even more the study.
Reviewer 2 Report
The in-text references do not follow the journal's standards, as they are in APA.
The introduction, line 77-105, is more relevant to the results section and even discussion, rather than introduction.
From line 202 to 250, we seem to be in a discussion phase.
In general, I think there are sections that have not been added and are shown in red: they need to be amended.
It is also important to readjust the text of the justification and align it with results, discussion and conclusion, in order to give a better understanding of the project, which is currently not very understandable in the way it is presented, as it constantly refers to other studies and returns to the current project. The text should be adjusted : 1) to what is to be achieved.
2) to the sample with which it is intended to act
3) to the studies already carried out
Separating each one into its phases to give a better understanding of the study.
Author Response
Reviewer 2:
The in-text references do not follow the journal’s standards, as they are in APA.
We thank Reviewer 1 for pointing out this error. References have been adjusted to this journal’s standards.
The introduction, line 77-105, is more relevant to the results section and even discussion, rather than introduction.
In the original manuscript version we have, lines 77-105 correspond to our literature review beginning with “These findings suggest new and innovative strategies …” in the middle of a paragraph, and ending with “on happiness and subjective well-being [40,41]” also in the middle of a paragraph. We are afraid we do not have the same configuration indicated by the reviewer. Anyhow, supposing we are referring to the same text, we agree that it could be perfectly relevant to the Results section and Discussion section. However, considering that we have no results so far, and that this part relates to the literature review covering the three approaches being studied, we also think that we could maintain this part in the Introduction section, unless Reviewer 2 maintains this suggestion.
From line 202 to 250, we seem to be in a discussion phase.
In general, I think there are sections that have not been added and are shown in red: they need to be amended.
Again, we were not able to locate the part indicated by Reviewer 2
It is also important to readjust the text of the justification and align it with results, discussion and conclusion, in order to give a better understandable in the way it is presented, as it constantly refers to other studies and returns to the current project. The text should be adjusted: 1) to what is to be achieved. 2) to the sample with which it is intended to act 3) to the studies already carried out.
Separating each one into its phases to give a better understanding of the study.
We also thank reviewer 1 for these suggestions. Discussion have been adjusted in order to address them.
Reviewer 3 Report
The intervention protocol appears well structured, supported by bibliographic references and innovative.
a) The research is relevant because it proposes methods of intervention that: - can be useful in the treatment of PTSD caused by CoViD-19 - and, being provided online, they allow people immediate access to care, without having to leave home and in compliance with the rules of social distancing.
b) Methodology The authors, in paragraph "2.7 Interventions": - with respect to TBCT, they describe the type of intervention and its objectives and indicate various studies that support its effectiveness; - with respect to MBHP, they describe the intervention and objectives without mentioning efficacy studies but it would be interesting to indicate some in relation to PTSD; - even with respect to the PPT, no studies that support its effectiveness are cited.
c) In the research, the authors underline the objective of evaluating the online effectiveness of these therapeutic approaches, in particular of TBT, with respect to the reduction of PTSD. It is not clear whether this approach also acts on the intrusive and dissociative symptoms and on the alterations of arousal and reactivity that characterize this disorder.
d) The conclusions are consistent with the arguments presented and are in agreement with the hypotheses, considering that it is an experimental protocol for which, at the moment, the results are not available.
e) References are appropriate.
f) The tables are simple and explanatory.
Author Response
Reviewer 3:
The intervention protocol appears well structured, supported by bibliographic references and innovative.
- The research is relevant because it proposes methods of intervention that: - can be useful in the treatment of PTSD caused by CoVID -19 and, being provided online, they allow people immediate access to care, without having to leave home and in compliance with the rules of social distancing.
- Methodology The authors, in paragraph “2.7 Interventions”: - with respect to TBCT, they describe the type of intervention and its objectives and indicate various studies that support its effectiveness; - with respect to MBHP, they describe the intervention and objectives without mentioning efficacy studies but it would be interesting to indicate some in relation to PTSD; - even with respect the PPT, no studies that support its effectiveness are cited.
We thank Reviewer 3 for this suggestion. We added one paragraph for MBHP and another for PPT citing their effectiveness.
- In the research, the authors underline the objective of evaluating the online effectiveness of these therapeutic approaches, in particular of TBCT, with respect to the reduction of PTSD. It is not clear whether this approach also on the intrusive and dissociative symptoms and on the alterations of arousal and reactivity that characterize this disorder.
The only existing study on TBCT for PTSD so far was conducted by our team (Ref). We did not address this issue explicitly in that study. Intrusive and dissociative symptoms seemed to respond to the treatment. However, as it was not a dismantling study, we are not sure about the mechanism responsible for these symptoms improvement.
- The conclusions are consistent with the arguments presented and are in agreement with the hypotheses, considering that it is an experimental protocol for which, at the moment, the results are not available.
- References are appropriate
- The tables are simple and explanatory
Round 2
Reviewer 2 Report
The research has achieved an achievement in introducing the topic of study, showing the objectives and the methodology to be applied, as well as the writing of the article in the results and discussion.
The subject is presented in a clean and orderly manner that gives rise to an understanding of both the state of the question and the study to be carried out.
I find the research interesting and publishable at this time.
I am grateful for the modifications made to the manuscript